# Physicians' attitude towards webinars and online education amid COVID-19 pandemic: When less is more

**Ismail Ibrahim Ismail**  [1]*, **Ahmed Abdelkarim** [2], **Jasem Y. Al-Hashel** [1,3]

**1** Department of Neurology, Ibn Sina Hospital, Kuwait City, Kuwait, **2** Department of Neuropsychiatry, Faculty of Medicine, Alexandria University, Alexandria, Egypt, **3** Department of Medicine, Faculty of Medicine, Health Sciences Centre, Kuwait University, Kuwait City, Kuwait

\* dr.ismail.ibrahim2012@gmail.com

## Abstract

### Background

Since the declaration of COVID-19 as a pandemic, all scientific medical activities were shifted to an online format, in the form of webinars, to maintain continuing medical education (CME). We aimed to assess physicians' attitude among different medical specialties towards this sudden and unexpected shift of traditional face-to-face meetings into webinars, and to suggest future recommendations.

### Methods

We conducted a cross-sectional, internet-based survey study using a 25-item question-naire, from November 1 and November 15, 2020. The survey was created and distributed to physicians from different medical and surgical specialties and from different countries via several social media platforms, using a snowball technique.

### Results

A total of 326 physicians responded; 165 (50.6%) were females, mean age of responders was 38.7 ± 7.5 years. The majority of responses (93.2%) came from Arab countries. Of them, 195 (59.8%) reported attending more webinars compared to the same period last year, with average of 3 per month. As regard to the general impression; 244 (74.8%) were "strongly satisfied" or "satisfied", with the most satisfaction for "training courses: by 268 (82.2%), and "International conferences" by 218 (66.9%). However, 246 respondents (75.5%) felt overwhelmed with the number and frequency of webinars during the pandemic, 171 (52.5%) reported attending less than 25% of webinars they are invited to, 205 (62.8%) disagreed that webinars can replace in-person meetings after the pandemic, and 239 (73.3%) agreed that online meetings need proper regulations.

**Data Availability Statement:** All relevant data are within the manuscript and its Supporting Information files.

**Funding:** The author(s) received no specific funding for this work.

**Competing interests:** The authors have declared that no competing interests exist.

## Conclusions

Webinars comprised a major avenue for education during COVID-19 pandemic, with initial general satisfaction among physicians. However, this paradigm shift was sudden and lacked proper regulations. Despite initial satisfaction, the majority of physicians felt overwhelmed with the number and frequency of webinars. Physicians' satisfaction is crucial in planning future educational activities, and considering that this current crisis will most likely have long lasting effects, webinars should be viewed as complementing traditional in-person methods, rather than replacement. In this study, we are suggesting recommendations to help future regulation of this change.

## Introduction

The world has been overtaken by coronavirus 2019 (COVID-19) pandemic since March 11, 2020. Since then, health care systems around the world have been struggling to contain this unprecedented public health crisis [1]. Most countries implemented lockdown and social distancing recommendations for fear of COVID-19 spread, and the majority of medical conferences and in-person scientific meetings were canceled. Many societies and educational institutes started using online-meeting applications to reach their relevant goals and needs, and the medical educational organizations and pharmaceutical companies were no exception [2]. The scientific activities were shifted to an online format, in the form of webinars, virtual conferences and online teaching courses. Webinars (web-based seminars) are one form of online communication tools that facilitate delivery of scientific information and sharing experiences among physicians. Thus, it best suited a time of movement restrictions and fear from a potentially fatal virus amid COVID-19 pandemic [3].

During the 6-month period following the pandemic, physicians have been invited to attend a lot of scientific webinars, while working to minimize the spread of COVID-19 outbreak. Webinars at this point were probably the only solution for continuing medical education (CME), however, months into the pandemic, their number and frequency has markedly increased. Physicians had difficulties in controlling their virtual agenda and struggled to participate in the extensive offer of webinars, teaching courses, medical representatives visits, board and committee meetings, in addition to telemedicine. It is not surprising that several physicians have felt "digital burnout" as suggested by some authors [4]. The ability of physicians to attend these online educational activities were compromised by several factors, and the lack of proper regulations as regard to the best teaching format, timing to conduct webinars, or methods for feedback, created several challenges amid COVID-19 crisis [5, 6].

Several studies have discussed these emerging challenges and disadvantages of online education amid COVID-19 pandemic [7, 8], however, there are limited data in the literature regarding physicians' perception and satisfaction towards this sudden and unexpected shift in medical education. We believe that physicians' satisfaction is crucial, not only for evaluation of the current practices, but also for planning of future strategies between in-person and virtual learning. In this survey study, we aim to assess physicians' attitude, mainly in The Middle East and North Africa (MENA) region, towards webinars in comparison to the traditional in-person meetings, and to suggest future recommendations to regulate this change in practice.

## Methods

### Participants

This cross-sectional, internet-based study, was conducted between November 1 and November 15, 2020. The study was approved by the Institutional Review Board Committee of Ministry of health of the state of Kuwait. The survey was developed and distributed to physicians of different specialties and from different countries via several social medical networks using a snowball technique. Answering the survey was considered an implied consent to participate in the study, and all answers were confidential. The study follows the American Association for Public Opinion Research (AAPOR) reporting guideline.

### Survey

A 25-item questionnaire was designed in English language, combining multiple choice and Likert response scale questions, with the option for respondents to provide further free text responses for some questions. A pilot study was conducted on a random sample of 25 physicians to determine the feasibility of using the questionnaire. Each author sent the questionnaire to a random of 10 physicians of their contacts, and 25 responded, which was considered a convenient sample. The survey was then modified as regard to formulation of the questions, formulation of response preferences, and the addition of certain demographic variables before distribution. The survey aimed to assess physicians' attitude towards shifting scientific activities to online formats amid COVID-19 pandemic, using a 5-point Likert scale of agreeableness ('strongly disagree', 'disagree', 'neutral', 'agree', and 'strongly agree') against certain questions. The self-administered survey was developed using "Google™ Forms", which require participants to sign-in into their Google accounts to prevent multiple entries from respondents.

The educational activities can be different among medical specialties, but they mainly include; general online webinars, teaching courses on specific subjects (e.g. Botox injection), International annual conferences, or pharmaceutically-sponsored meetings discussing specific therapies.

The survey collected the following data from three domains: (1) demographic variables (age, gender, country of practice, specialty, years of experience, working role during the pandemic). (2) webinars-related variables (attendance, number of attendances per month, comparison to last year attendance, role in webinars, percentage of webinars attended in relation to the number of invitations). (3) physicians' attitude towards webinars (general impression on shifting scientific meetings to online format, specific impression on shifting international conferences/ teaching courses versus pharmaceutically-sponsored meetings to online format, impression on the scientific content of international conferences versus pharmaceutically-sponsored meetings in online format, types of online meetings respondents prefer to attend, types of online meetings respondents reject to attend, factors determining attendance or rejection to attend online meetings, agreement on online meetings replacing in-person visits, feeling overwhelmed with the number and frequency of online meetings, agreement on the need for further regulations). In reporting the results, the data from the two columns of "strongly agree" and "agree"/ "strongly satisfied" and "satisfied" were combined and the data for "strongly disagree" and "disagree"/ "strongly dissatisfied" and "dissatisfied" were combined as well. Participants were asked at the end of the survey to give their opinion and to suggest recommendations for educational organizations, pharmaceutical companies, and policy makers, to improve physicians' satisfaction beyond the COVID-19 pandemic.

The full survey instrument can be viewed as an appendix to this article (S1 Appendix).

The survey link was created and initially posted on the personal social media accounts of the primary investigators and on several medical groups on Facebook. Physicians were

encouraged to recruit other colleagues by resending the link to their contacts through a "snow-ball technique". The link was posted only once on each medical group, and twice on the personal accounts of the investigators. The survey continued until there was no more responses for 1 day. The survey was closed at 23:59, November 15, 2020.

### Statistical analysis

Data were analyzed using SPSS statistical software version 25.0 (Armonk, NY: IBM Corp). The Kolmogorov- Smirnov non-parametric test was used to verify the normality of distribution of variables. Comparisons between groups for categorical variables were assessed using Chi-square test (Fisher's Exact or Monte Carlo correction). Student t-test was used to compare two groups for normally distributed quantitative variables while Mann Whitney test was used to compare between two groups for not normally distributed quantitative variables. Significance of the obtained results was judged at the 5% level.

## Results

In this survey, a total of 326 physicians responded to the online survey; 165 (50.6%) females, and 161 (49.4%) were males, with a mean age of 38.7 ± 7.5 years for all responders. Mean years of experience for the cohort was 11.8 ± 7.01 years. The respondents were mostly from Arab countries (93.2%); Egypt; 142 (43.6%), Kuwait; 122 (37.4%), UAE; 22 (6.7%), KSA; 13 (4%) and 27 (8.3%) from other countries. The response rate cannot be calculated in this study. During the pandemic, 172 physicians (52.8%) continued to practice their usual work, 80 (24.5%) were frontline healthcare worker, 38 (11.7%) partially worked, 24 (7.4%) practiced through "telemedicine services", while 12 (3.7%) didn't work during the pandemic. Physicians' characteristics and their responses to the survey questions are summarized in **Table 1**. Of the respondents, the commonest specialties were neurology by 75 physicians (23%), psychiatry by 40 (12.3%), family medicine by 38 (11.6%), general surgery by 20 (6.1%), critical care by 17 (5.2%), internal medicine by 16 (4.9%), radiology by 16 (4.9%), and 104 physicians (31.9%) from other specialties, as summarized in **Table 2**.

The majority of physicians; 318 (97.5%) attended webinars or online meetings during the first 6 months of the pandemic, with 195 (59.8%) reported attending more webinars compared to the same period last year. The average median was 3 (0–20) webinars per month, compared to 0 (0–1) in the past year. More than half of the respondents; 171 (52.5%) reported attending less than 25% of webinars they are invited to, 72 (22.1%) attended 25–50%, 55 (16.9%) attended 50–75%, while only 28 (8.6%) reported attending more than 75%. Most of the respondents; 270 (82.8%) were "*mostly attendees*", 41 (12.6%) were "*equally speakers and attendees*", and 15 (4.6%) were "*mostly speakers*".

As regard to the general impression on shifting scientific meetings to online format, 244 (74.8%) were "strongly satisfied" or "satisfied", 49 (15%) were neutral, while 33 (10.1%) were "strongly dissatisfied" or "dissatisfied". Almost the same results were obtained when asked about their impression on shifting international conferences and teaching courses to online format; 243 (74.5%) were "strongly satisfied" or "satisfied", 42 (12.9%) were neutral, and 41 (12.5%) were "strongly dissatisfied" or "dissatisfied". However, these results differed when asked about shifting pharmaceutically-sponsored meetings; only 135 (41.4%) were "strongly satisfied" or "satisfied", 90 (27.6%) were neutral, while 101 (30.9%) were "strongly dissatisfied" or "dissatisfied".

As regard to impression on the scientific content of the international conferences in online format; 267 (81.9%) were "strongly satisfied" or "satisfied", 44 (13.5%) were neutral, and 15 (4.6%) were "strongly dissatisfied" or "dissatisfied". These results also changed when asked

**Table 1. Distribution of the survey respondents according to different parameters (n = 326).**

| | N (%) |
|---|---|
| **Age (years)** | |
| Mean ± SD. | 38.7 ± 7.5 |
| Median (Min.–Max.) | 37 (25–61) |
| **Gender** | |
| Male | 161 (49.4%) |
| Female | 165 (50.6%) |
| **Country of practice** | |
| Egypt | 142 (43.6%) |
| Kuwait | 122 (37.4%) |
| UAE | 22 (6.7%) |
| KSA | 13 (4%) |
| UK | 8 (2.5%) |
| USA | 7 (2.1%) |
| Germany | 4 (1.2%) |
| India | 3 (0.9%) |
| Oman | 3 (0.9%) |
| Syria | 2 (0.6%) |
| **Years of practice in your specialty** | |
| Mean ± SD. | 11.8 ± 7.01 |
| Median (Min.–Max.) | 11 (1–36) |
| **How do you describe your role amid COVID-19 pandemic?** | |
| Frontline healthcare worker | 80 (24.5%) |
| I continued to practice my usual work during the pandemic | 172 (52.8%) |
| I didn't work during the pandemic | 12 (3.7%) |
| I partially worked during the pandemic | 38 (11.7%) |
| I practiced through "telemedicine" services | 24 (7.4%) |
| **Have you attended webinars or online meetings during the past 6 months?** | |
| No | 8 (2.5%) |
| Yes | 318 (97.5) |
| **If yes, how many webinars or online meetings, in average, have you attended per month during the past 6 months?** | |
| Median (Min.–Max.) | 3 (0–20) |
| **How many webinars or online meetings, in average, have you attended per month last year?** | |
| Median (Min.–Max.) | 0 (0–1) |
| **In comparison to the last year, have you attended more meetings and conferences during the past 6 months?** | |
| No | 110 (33.7%) |
| Maybe | 21 (6.4%) |
| Yes | 195 (59.8%) |
| **What is your role in webinars and online meetings?** | |
| Equally speaker and attendee | 41 (12.6%) |
| Mostly attendee | 270 (82.8%) |
| Mostly speaker | 15 (4.6%) |
| **In average, what is the percentage of webinars or online meetings you attend in comparison to the number you are invited to?** | |
| Less than 25% | 171 (52.5%) |
| 25–50% | 72 (22.1%) |
| 50–75% | 55 (16.9%) |
| More than 75% | 28 (8.6%) |

**Table 2. Distribution of the studied cases according to the different specialties (n = 326).**

| Specialty | N (%) |
|---|---|
| Anesthesia | 9 (2.8%) |
| Cardiology | 7 (2.1%) |
| Critical care | 17 (5.2%) |
| Dermatology | 7 (2.1%) |
| Endocrinology | 3 (0.9%) |
| Family medicine | 38 (11.6%) |
| General surgery | 20 (6.1%) |
| Haematology | 2 (0.6%) |
| Internal medicine | 16 (4.9%) |
| Neurology | 75 (23%) |
| Neurosurgery | 12 (3.7%) |
| Nutrition | 5 (1.5%) |
| OB/GYN | 2 (0.6%) |
| Oncology | 4 (1.2%) |
| Ophthalmology | 13 (4%) |
| Orthopedic surgery | 4 (1.2%) |
| Otolaryngology | 5 (1.5%) |
| Pain management | 1 (0.3%) |
| Palliative care | 1 (0.3%) |
| Pediatric surgery | 1 (0.3%) |
| Pediatrics | 11 (3.4%) |
| Physical medicine | 4 (1.2%) |
| Plastic surgery | 4 (1.2%) |
| Psychiatry | 40 (12.3%) |
| Public health | 1 (0.3%) |
| Pulmonology | 3 (0.9%) |
| Radiology | 16 (4.9%) |
| Rheumatology | 5 (1.5%) |

about the scientific content of the pharmaceutically-sponsored meetings; 120 (36.8%) were "strongly satisfied" or "satisfied", 86 (26.4%) were neutral, and 120 (36.8%) were "strongly dissatisfied" or "dissatisfied". The commonest types of meetings respondents preferred to attend were training courses by 268 (82.2%), international conferences by 218 (66.9%), local/regional conferences by 167 (51.2%), and the least was commercial/ pharmaceutically-sponsored meetings by 29 (8.9%) respondent. Findings are illustrated in **Table 3**, and **Fig 1**.

The commonest reported factors for determining attendance were; scientific content by 297 (91.1%), CME certification by 164 (50.3%), speaker's name and experience by 158 (48.5%), annual international conferences by 156 (47.9%), and personal relations with the inviting party by 61 (18.7%) respondent. The commonest reported factor for declining to attend was the timing of the meeting by 284 (87.1%), followed by the scientific content by 228 (69.9%), lack of CME certification by 115 (35.3%), personal factors by 85 (26.1%), speakers in the meeting by 75 (23%), and lack of personal relations with the inviting party by 50 (15.3%) respondent.

When asked if webinars can replace in-person meetings after the pandemic; 205 respondents (62.8%) reported "strongly disagree" or "disagree", in comparison to 105 (32.2%) who reported "strongly agree" or "agree". Moreover, 246 (75.5%) agreed or strongly agreed that

**Table 3. Distribution of survey responses according to satisfaction with different webinars and online meetings (n = 326).**

| | Strongly dissatisfied | Dissatisfied | Satisfied | Strongly satisfied | $X^2$ | p |
|---|---|---|---|---|---|---|
| Since the beginning of COVID-19 pandemic, what is your general impression on shifting scientific meetings to webinars and online meetings? | 4 (1.2%) | 29 (8.9%) | 183 (56.1%) | 61 (18.7%) | 272.704* | <0.001* |
| What is your impression on shifting international conferences and teaching courses to webinars and online meetings? | 2 (0.6%) | 39 (12%) | 158 (48.5%) | 85 (26.1%) | 190.845* | <0.001* |
| What is your impression on shifting pharmaceutically-sponsored meetings to webinars and online meetings? | 4 (1.2%) | 97 (29.8%) | 69 (21.2%) | 66 (20.2%) | 78.271* | <0.001* |
| In general, what is your impression on the scientific content of the International conferences as webinars and online meetings? | 1 (0.3%) | 14 (4.3%) | 200 (61.3%) | 67 (20.6%) | 391.703* | <0.001* |
| In general, what is your impression on the scientific content of the pharmaceutically-sponsored meetings as webinars and online meetings? | 7 (2.1%) | 113 (34.7%) | 91 (27.9%) | 29 (8.9%) | 125.667* | <0.001* |

$\chi^2$: **Chi square for Goodness of fit.**

*: **Statistically significant at p $\leq$ 0.05.**

they felt overwhelmed with the number and frequency of webinars during the pandemic, with 239 (73.3%) agreed or strongly agreed that webinars and online meetings need further regulations. Findings are illustrated in **Table 4**, and **Fig 2**.

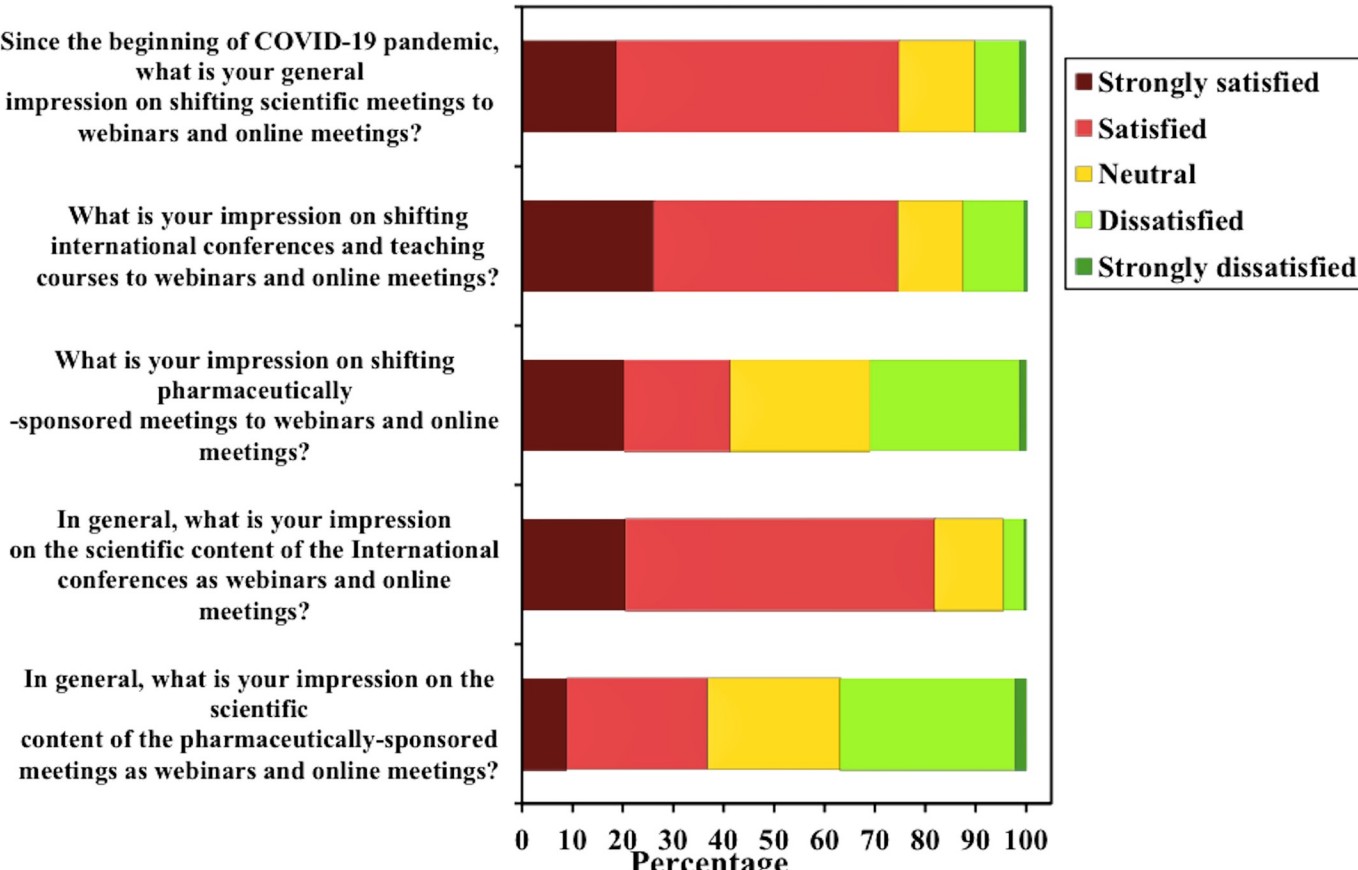

**Fig 1. Stacked bar chart of Likert questions.** The figure shows 5 bar charts describing the impression of physicians towards shifting scientific activities into online format, the impression differences between international conferences and pharmaceutically-sponsored activities, and the perception of the scientific content of these activities during the pandemic.

**Table 4. Distribution of the survey responses according to agreeableness to different items (n = 326).**

| | Strongly disagree | Disagree | Agree | Strongly agree | $X^2$ | p |
|---|---|---|---|---|---|---|
| Do you agree that webinars and online meetings can replace in-person meetings after the pandemic? | 110 (33.7%) | 95 (29.1%) | 78 (23.9%) | 27 (8.3%) | 50.490* | <0.001* |
| Have you felt overwhelmed with the number and frequency of webinars and online meetings during the pandemic? | 0 (0%) | 31 (9.5%) | 123 (37.7%) | 123 (37.7%) | 61.112* | <0.001* |
| Do you agree that webinars and online meetings need further regulations? | 0 (0%) | 14 (4.3%) | 127 (39%) | 112 (34.4%) | 89.320* | <0.001* |

$\chi^2$: Chi square for Goodness of fit.

*: Statistically significant at $p \leq 0.05$.

A statistical comparison was performed between satisfied and non-satisfied groups, and there were no significant differences as regard to age (p = 0.601), gender (p = 0.593), years of clinical practice (p = 0.620), and role amid the pandemic (p = 0.563). However, there was a statistically significant difference regarding the role during online meetings, where "*mostly speakers*" reported more satisfaction compared to "*mostly attendees*" (p = <0.001). Findings are summarized in Table 5.

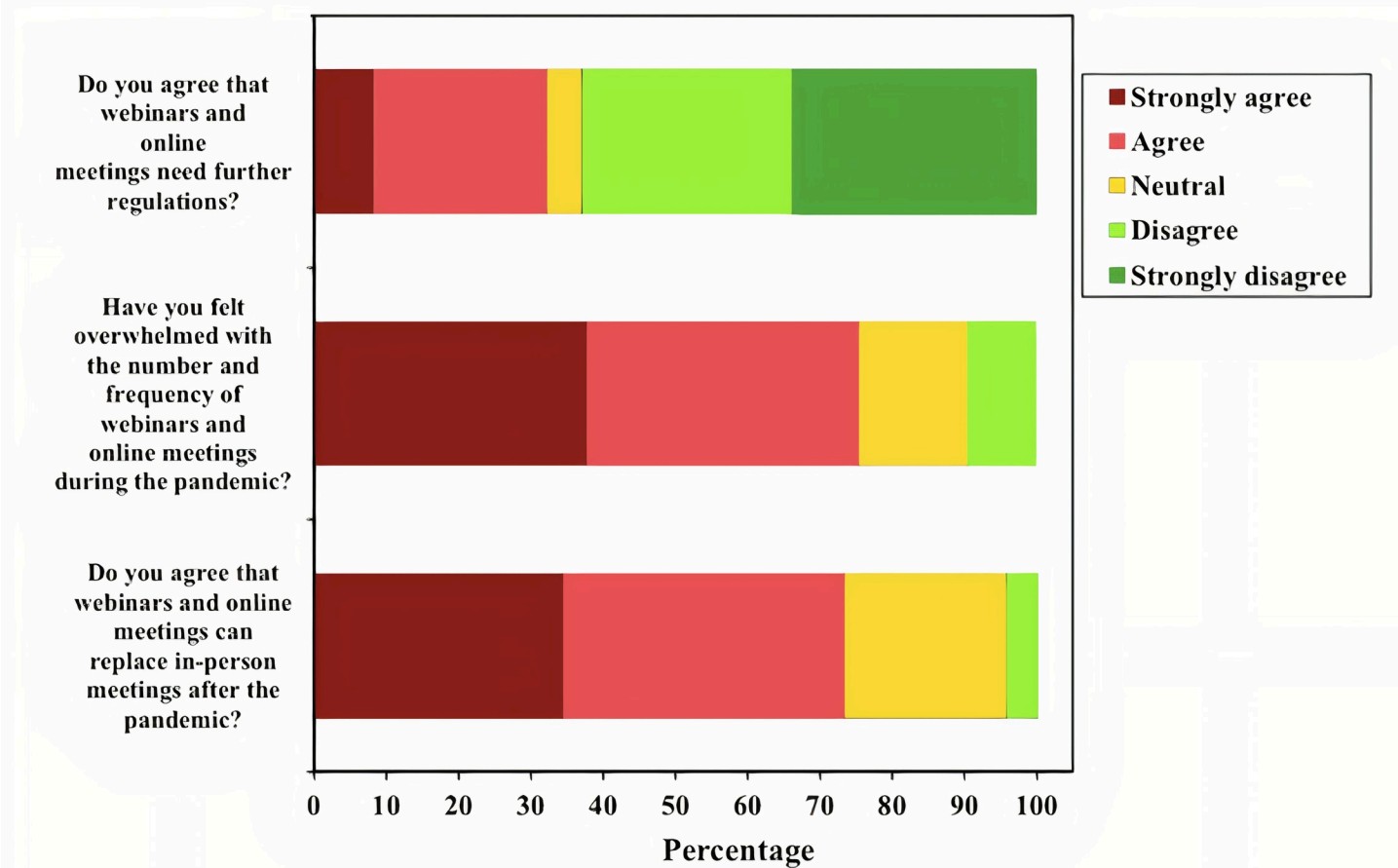

**Fig 2. Stacked bar chart of Likert questions.** The figure shows 3 bar charts describing the questions regarding being overwhelmed with webinars, the need for further regulations, and if webinars can replace traditional in-person methods after the pandemic.

**Table 5. Comparison between satisfied and non-satisfied groups on shifting scientific meetings into webinars with different parameters (n = 277).**

| | General impression on shifting scientific meetings into webinars | | Test of Sig. | p |
|---|---|---|---|---|
| | Satisfied (n = 33) | Dissatisfied (n = 244) | | |
| **Gender** | | | | |
| Male | 15 (45.5%) | 123 (50.4%) | $\chi^2$ = | 0.593 |
| Female | 18 (54.5%) | 121 (49.6%) | 0.286 | |
| **Age (years)** | | | | |
| Mean ± SD. | 39.5 ± 7.7 | 38.7 ± 7.5 | t = | 0.601 |
| Median (Min.–Max.) | 38 (26–59) | 37.5 (25–61) | 0.523 | |
| **Years of practice** | | | | |
| Mean ± SD. | 12.3 ± 6.7 | 11.8 ± 7.1 | U = | 0.620 |
| Median (Min.–Max.) | 12 (1–27) | 11 (1–36) | 3812.0 | |
| **How do you describe your role amid COVID-19 pandemic?** | | | | |
| Frontline healthcare worker | 8 (24.2%) | 58 (23.8%) | $\chi^2$ = | MCp = |
| I continued to practice my usual work during the pandemic | 16 (48.5%) | 139 (57%) | | |
| I didn't work during the pandemic | 1 (3%) | 4 (1.6%) | | |
| I partially worked during the pandemic | 6 (18.2%) | 26 (10.7%) | 2.821 | 0.563 |
| I practiced through "telemedicine" services | 2 (6.1%) | 17 (7%) | | |
| **What is your role in webinars and online meetings?** | | | | |
| Equally speaker and attendee | 9 (27.3%) | 19 (7.8%) | $\chi^2$ = | MCp = |
| Mostly attendee | 19 (57.6%) | 216 (88.5%) | 18.142* | <0.001* |
| Mostly speaker | 5 (15.2%) | 9 (3.7%) | | |

$\chi^2$: **Chi square test**, MC: **Monte Carlo**, t: **Student t-test**, U: **Mann Whitney test**.

p: p value for association between different categories.

*: Statistically significant at p ≤ 0.05.

## Discussion

The COVID-19 pandemic has caused a major disruption to the conventional medical education across the world, and webinars were used in an attempt to maintain teaching and learning. This phenomenal shift of concepts has led to an increase in webinar usage in 2020 compared to the same period in 2019, by more than 300% in one study [9] and up to 3250% in another [10]. This is not the first time when traditional educational activities are suspended in a time of a major crisis. SARS coronavirus (SARS-CoV) and H1N1 Flu outbreaks also negatively impacted educational activities in a large number of countries around the globe, enforcing "emergency remote teaching" as a feasible option [11]. However, the current circumstances are unique and different in its global magnitude and the more likely long-lasting effects after the pandemic.

In this study, nearly two-thirds of physicians attended more meetings during the first 6 months of the pandemic compared to the same period last year, and the majority reported initial satisfaction. This satisfaction was higher with "International conferences" and "teaching courses" compared to pharmaceutically-sponsored meetings (74.5% vs 41.4%, p = <0.001). Moreover, the satisfaction with the scientific content was also higher between the aforementioned two types of meetings (81.9% vs 36.8%, p = <0.001). This interesting disparity in satisfaction might be attributed to the higher number, frequency, and overlapping activities of pharmaceutically-sponsored meetings, probable perceived biases, or due to dominance of early COVID-19-related topics at that time. Moreover, International conferences are annual

events, that usually lack commercial biases, and are eligible for higher CME accreditation, than pharmaceutically-sponsored events.

However, the majority (75.5%) reported feeling overwhelmed with the frequency of online meetings during this period, more than half (52.5%) attended less than 25% of webinars they are invited to, and (73.3%) felt the need for proper regulations of this practice. Furthermore, 62.8% disagreed that online format can replace in-person meetings afterwards.

These findings were in line with other surveys by Figueroa et al. [12] and Al-Ahmari et al. [13], where around 60% to 70% of physicians believed that webinars should not replace face-to-face traditional teaching after the pandemic, respectively. Moreover, similar findings were found among medical students in a study that included 13 medical schools, where 54.8% disagreed that e-learning could be used for clinical teaching [14].

Interestingly, similar findings extended to physicians' attitude towards online consultations with patients. In a survey study of 140 physicians from Lebanon, the vast majority of responders disagreed, or remained undecided, as to whether "telemedicine" can replace face-to-face consultations [15].

Respondents reported several reasons for dissatisfaction including: the overwhelming number and frequency, lack of scheduling, increase screen time exposure especially with shifting practice to telemedicine, technical difficulties, increased stress from either attending too many or too little webinars for fear of missing out on educational activities, in addition to disturbance of work/life balance, as most webinars occur during weekends or after working hours. In a study evaluating challenges to online medical education during the COVID-19 pandemic in Saudi Arabia [5], poor communication (59%), lack of assessment (57.5%), technology-related issues (56.5%), online inexperience (55%), pandemic-related anxiety or stress (48%), time management (35%), and technophobia (17%) were the commonly reported factors. In the same study, 62.5% of respondents preferred combining online with traditional face-to-face education, 25.5% preferred traditional face-to-face, and only 12.0% preferred online education alone.

The "*Anywhere-Anytime*" feature of webinars were found to be especially beneficial at the beginning of COVID-19 pandemic. However, this same feature was reported to be one the most relevant reasons for dissatisfaction. Physicians' satisfaction largely affects the willingness to continuously use the platform. Online education advantages are numerous including time and location flexibility, being less costly, possibility of self-paced education, and accommodation of a wider audience. However, they also pose several disadvantages such as technical difficulties, less communication with attendees, slow response time with low Internet connection, increase distraction during webinars, limited feedback and assessment, and absence of socialization in comparison to the traditional ways [16, 17]. Moreover, it is well known that high screen time exposure can be linked to increase stress, sleep disturbance, fatigue, anxiety and depression [18, 19]. Furthermore, little evidence has been presented regarding the efficiency and efficacy of online education compared to traditional methods, especially with the huge differences between countries in digital advancement [20], in addition to difficulties to replicate in-person clinical bedside experiences virtually.

Higher satisfaction among speakers in comparison to attendees can be understood in the light of several studies, showing that acceptance of change is affected primarily by the degree to which individuals are involved in the change process. In other words, participation in implementing the various aspects of online education leads to significantly less resistance to this change [21].

## Recommendations

William Osler once said, "Medicine is learned by the bedside and not in the classroom" [22].

With this in mind, and though attending webinars is optional, it became a major avenue to maintain the provision of medical education remotely for the time being. The debate about their benefits and limitations is still ongoing, and despite earlier attempts to regulate online education in general [23, 24], the majority of respondents agreed that regulations are needed, as part of preparation for future pandemics or other disruptions to medical education. The following recommendations are suggested based on our survey findings, and a review of the current literature [25–34].

1-Webinars should be regulated through the local/ regional scientific organizations to arrange for the timing, frequency, and topics, in order to avoid repetition and overlapping activities and to help limiting screen-time exposure. In numerous everyday domains, it has been demonstrated that increasing the number of options can lead to a choice paralysis and decrease satisfaction; the so-called "*paradox of choice*" [25]. A predetermined monthly schedule can be helpful in avoiding overwhelm and burnout.

2- Inconvenient timing was found to be the most important concern in our survey. Most respondents preferred avoiding working hours and weekends, to avoid disruption of work/life balance, however, time zone differences still remain an important challenge for International events. Studies have shown that the best time to hold a webinar is mid-week, with a strong preference for Tuesday and Wednesday [26].

3- Research in the field of "brain-based learning" has resulted in some useful "learner-centered" techniques for planning the content of webinars to maintain high attention and concentration levels: e.g. sessions should be 20 minutes or less, followed by 5 minutes of discussion, stimulation of as many of the learner's senses as possible using images, question-response modules, games, or role-playing, and to provide timely assessment and feedback [26].

4- Webinars should be concise, well-structured, and having clear objectives. Most general webinars should not be longer than one hour in total, whereas webinars presenting highly technical content, or webinars for training purposes, can be up to a maximum of 1.5 hours [27, 28].

5- Speakers should be highly qualified, well known, and respected in their field. If more than one speaker is required, those with opposing views can be chosen, to make the discussion livelier.

6-The scientific content should be novel, updated, and non-biased. The virtual experience should be enriched with multimodal learning elements including videos, images, and audio files, rather than merely presenting slides, in order to attract and engage more audience. Moreover, emphasis on social interaction among participants is crucial [29, 30].

7- The "two-way interaction" of teaching must be implemented. Knowledge assessment can be made prior to and after webinars, to ensure better understanding of the subjects. An interactive Q&A during live sessions is also beneficial [31, 32].

8—CME accreditation is important to ensure high quality of webinars, as physicians usually require a specified number of credits annually to maintain medical licenses.

9- Technical issues should be addressed, in order to ensure digital equity. Internet connection problems is a difficulty that needs to be anticipated, as not all attendees/speakers have access to the same amount of bandwidth [33, 34].

10-A recorded version of the lectures can be kept for a specific period of time, to allow for better self-paced learning.

## Limitations

Our study has some limitations which should be stated. First, being a cross-sectional survey study carries a potential for certain biases. Moreover, respondents self-selected themselves into the study via social media, which may be prone to selection bias, with nearly third of respondents from neurology and psychiatry. However, the other two-thirds came from a representative sample of different medical and surgical specialties. Furthermore, the majority of respondents were from physicians working in MENA countries, which could limit the generalizability of our results to other geographical regions. The results of this survey should be interpreted as descriptive, aiming to assess attitude of a defined population, towards a specific target, during a specific period of time. Second, our study was not designed to allow for sub-analysis of physicians' satisfaction according to different medical or surgical specialties, and further research is needed to understand satisfaction can differ between specialties. Finally, the survey was not validated, due to time constraints during the COVID-19 crisis, and the need to assess the outcome in a timely manner, which might limit the reliability in assessing physicians' attitude. However, we conducted a pilot study on a random sample of 25 physicians before distributing the survey, to determine its feasibility.

## Conclusion

The impact of COVID-19 pandemic on medical education has been significant all over the world. Webinars and online meetings provided the best avenue for teaching during this global crisis. Despite its several advantages, this paradigm shift came suddenly and without proper preparation from the educational organizations, and lacked proper regulations. Physicians in our study showed initial satisfaction with this shift, with higher satisfaction levels for "International conferences" and "teaching courses" compared to "pharmaceutically-sponsored meeting". However, the majority felt overwhelmed with the number and frequency of webinars, more than half attend less than quarter of webinars they were invited to, and most respondents thought it needs proper regulations. Moreover, two-thirds of physicians disagreed that webinars can replace in-person meetings. Considering that this current crisis will most likely have long lasting effects, webinars should be viewed as complementing traditional in-person methods, rather than replacement, while aiming for normalcy.

The findings and recommendations from this study highlight the importance of physicians' satisfaction in future planning of long-term strategies for online medical education. Further general and specialty-specific studies are needed to modify and refine proper online educational tools.

## Supporting information

**S1 Appendix. Survey questions.**
(DOCX)

## Acknowledgments

We would like to express our thanks and gratitude to our colleagues who participated by answering this survey study.

## Author Contributions

**Conceptualization:** Ismail Ibrahim Ismail.

**Data curation:** Ismail Ibrahim Ismail, Ahmed Abdelkarim, Jasem Y. Al-Hashel.

**Formal analysis:** Ismail Ibrahim Ismail, Ahmed Abdelkarim, Jasem Y. Al-Hashel.

**Investigation:** Ismail Ibrahim Ismail, Ahmed Abdelkarim, Jasem Y. Al-Hashel.

**Methodology:** Ismail Ibrahim Ismail, Ahmed Abdelkarim, Jasem Y. Al-Hashel.

**Project administration:** Ismail Ibrahim Ismail.

**Supervision:** Ismail Ibrahim Ismail, Ahmed Abdelkarim, Jasem Y. Al-Hashel.

**Validation:** Ismail Ibrahim Ismail, Ahmed Abdelkarim, Jasem Y. Al-Hashel.

**Visualization:** Ismail Ibrahim Ismail, Ahmed Abdelkarim, Jasem Y. Al-Hashel.

**Writing – original draft:** Ismail Ibrahim Ismail.

**Writing – review & editing:** Ismail Ibrahim Ismail.

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
