## [Decision Letter · Decision Letter 0]

1 Feb 2021

PONE-D-20-39984

Physicians' attitude towards webinars and online education amid COVID-19 pandemic: When Less is More

PLOS ONE

Dear Dr. Ismail,

Thank you for submitting your manuscript to PLOS ONE. After careful consideration, we feel that it has merit but does not fully meet PLOS ONE’s publication criteria as it currently stands. Therefore, we invite you to submit a revised version of the manuscript that addresses the points raised during the review process.

I wish to be sincere and say that of five reviewers, four presented significant issues, questioning whether the study as performed, deserves to be accepted for publication.  I would like to emphasize one criticism in a positive way.  The overwhelming majority of the responders were from Arab countries. Consider transforming your study into one that relates to this specific population. 

We look forward to receiving your revised manuscript.

Kind regards,

Itamar Ashkenazi

Academic Editor

PLOS ONE

Journal Requirements:

3. Please include your tables as part of your main manuscript and remove the individual files. Please note that supplementary tables (should remain/ be uploaded) as separate "supporting information" files-

4. Please upload a copy of Figure 3, to which you refer in your text on page 17. If the figure is no longer to be included as part of the submission please remove all reference to it within the text.

5. Please ensure that you refer to Figure 1 in your text as, if accepted, production will need this reference to link the reader to the figure.

Reviewers' comments:

Reviewer's Responses to Questions

**Comments to the Author**

1. Is the manuscript technically sound, and do the data support the conclusions?

Reviewer #1: Partly

Reviewer #2: Yes

Reviewer #3: Partly

Reviewer #4: Partly

Reviewer #5: Partly

2. Has the statistical analysis been performed appropriately and rigorously? 

Reviewer #1: Yes

Reviewer #2: Yes

Reviewer #3: I Don't Know

Reviewer #4: Yes

Reviewer #5: Yes

3. Have the authors made all data underlying the findings in their manuscript fully available?

Reviewer #1: Yes

Reviewer #2: Yes

Reviewer #3: Yes

Reviewer #4: No

Reviewer #5: Yes

4. Is the manuscript presented in an intelligible fashion and written in standard English?

Reviewer #1: No

Reviewer #2: Yes

Reviewer #3: No

Reviewer #4: No

Reviewer #5: Yes

5. Review Comments to the Author

Reviewer #1: Presentation

I appreciate the amount of work the authors have undertaken in this endeavor and have a few suggestions and clarifications that I hope may add to the value of this submitted manuscript.

1. Abstract

• The title of the article is clear and describes the focus of the study.

• The abstract is well structured and describes the study.

2. Introduction, the rationale and overall impact of the review are not clear. Some parts of the introduction need greater references to the evidence base.

3. Methods, the method section is in detail but there is no flow and many relevant pieces of information are missing:

• For international readers, and for readers who may wish to cite this study, it might be helpful for the authors to provide a brief overview of training activities at this particular institution to better facilitate an understanding of the results and discussion, especially as it applies to other institutions.

• Sampling technique and sampling size was not mentioned

• What were the baseline characteristics of the physicians such as qualification, specialties etc.

• How the consent from the physicians was taken?

• Who designed the survey?

• How the validity of the instrument was checked and what was the reliability score?

• It seems the snowball technique was not effectively used. Authors need to review the technique and explain in detail such as how they contact the first group, from where they got other contacts, who provided further contacts, and so on. In the snowball technique researcher asks for contacts and then asks contacts for participation. I am really curious about the process; how many reminders were sent and what was the logic for closing the link on November 15?

• The significance level should be checked.

4. Results & discussion, well written but the discussion needs more relevant referencing.

5. Conclusion, is relevant but due to major problems with the methods, it is really difficult to comment. In the conclusion, it would be useful to put the ideas in the context of what has been found. What is novel from this study? Strangely the validity of the tool was not done. Also, I am confused with the reason of “time constraint”. This needs further explanation.

6. Recommendations, are too long and most of them are not based on the results. Please review it.

7. References, Vancouver style was not followed.

8. It is unclear the additional value of this manuscript contributes to the literature beyond just a description of an attempt at investigating the physicians’ attitude toward webinars. The authors may need to highlight the values/implications of the current study (In the Introduction sections).

I also suggest careful editing of the manuscript to ensure clarity.

Reviewer #2: Dear authors

Thank you for submitting your manuscript to the PLOS ONE journal. Your study is a cross-sectional study using a questionnaire. I think this manuscript is suitable for a letter to the editor. There is no novelty in the idea or the design.

Reviewer #3: The overall outcomes of this study are interesting and highlights the feelings of physicians on using online methods for CME etc. However, I feel that in parts it is superficial and needs re-written.

To me there needs to be greater clarity in how the results are being presented and discussed, with more cited comparisons (where appropriate) of the specific points.

I also think that some interesting points that the survey has identified have been overlooked and would have made for further discussion.

In the main the recommendations are good but follow a common-sense approach that many institutions already expect.

Given the International nature of medical education these days it is hard to select a time that suits all possible attendees around the world. The authors recommendation number 1 is therefore particularly difficult to address, unless it is time zone specific.

There is a good argument for having an institutional body to coordinate local CME activities, but again it is extremely difficult to coordinate if it is a webinar being presented by a specific professional society in another country. In the end it should be up to the physician to decide which webinar they want to attend or not.

Specific points

Abstract - I think this needs to be re-written on the basis of comments on the full paper.

Introduction -

Page 10. Line 19. What ‘proper regulation’ are the authors alluding to at this point in the paper?

Page 10. Line 20. ‘several challenges and disadvantages have been reported by physicians.’ Is there a reference to this report?

Page 11. Line 10. Details of the ‘snowball technique’ should be provided.

Page 11. Line 17. How was a random sample of 25 physicians achieved? It is not clear how that was truly random

It is not clear why a distinction was drawn between international events and pharmaceutical sponsored events. this should be explained.

Statistical analysis. I don’t think there is a need to include the dates that analysis was done.

Kolmogorov- Smirnov – should include ‘non-parametric test’

Results.

Why mention only female respondents and their age range? I think male respondents and their age range should also be included. That would make an interesting comparison.

The way that the sentences 1 and 2 are written, suggests that the mean years of experience relates to the female respondents.

Page 14. Lines 9 and 10. ’Most of the respondents; 270 (82.8%) were mostly attendees’ or ‘15 (4.6%) were mostly speakers’ – using the term ‘mostly’ makes this unclear.

Page 15. The figures presented in the first 2 paragraphs, indicate that respondents answered twice or more. Perhaps these results should be presented as a ranked order of preferences.

Page 15. Line 19. ‘that webinars and online meetings need further regulations’. If they need further regulation, what were the initial regulations?

Discussion

Page 16. Lines 2 & 3. Again, results are being described as ‘mostly’.

Page 16. Line 20. ‘Moreover, the satisfaction with the scientific content was also higher between them (81.9% vs 36.8%)’. This statement needs to clarify between which groups.

Page 17. line 4. The authors again mention the need for further regulations, but there has been no mention of existing regulations.

Page 18 line 1. It is not clear what the authors mean by poor communication, slow response time and increased distraction. I think these should be clarified.

Page 18. Line 5. Suggestion – ‘Furthermore, little evidence has been presented…..’

Recommendations.

Page 19. Line 6. ‘Studies have shown……’ Needs a reference here.

Page 19. Line 8. ‘Research in the field…..’ Needs a reference here.

Page 20. Line 1. ‘The virtual experience should be enriched…’ It would be helpful if the authors gave examples of enrichment.

Conclusion.

The authors confuse the issue here. One the one hand they say ‘lacked proper regulation’ and on the other they say ‘the need for further regulation’, implying that there was some form of regulation in the first place.

Reviewer #4: Dear respected Authors,

Thanks for your efforts on doing research and trying to contribute to the field of continuing medical education.

I have found your research question interesting and read your manuscript with interest.

Nonetheless, I have a few concerns regarding the manuscript which once addressed I believe would make the manuscript better.

Abstract: Overall fine. It is better to rephrase “continuous medical education” to “continuing medical education (CME)”.

Introduction:

The claims in the second paragraph should supported with relevant literature.

In the purpose statement it has been stated that you aimed to assess “physicians’ attitude” which requires to use a valid reliable scale. Your aim could be defining the “physicians’ reactions” with the survey you developed.

Method:

Your sampling method sounds more like convenience sampling rather than snowball sampling. I believe one of the major problems with your manuscript is the distribution of the survey questionnaire relatively uncontrolled way. Although you have distributed your survey questionnaire via multiple social media platforms, you have received vast majority of the responses from Middle East and North Africa region? Could it because of the survey questionnaire language? Was it in English or Arabic? This information should be included in the text.

What happened after the pilot study of the survey? Were there any changes or additions to the questionnaire?

Google Forms statement repeated multiple times in the manuscript which seems unnecessary.

While it is a good measure to ask sign-in into Google accounts to avoid multiple responses from participants, this is also a limitation for “possible participants” who are not regularly using Google accounts or reluctant to log in. I think it would be better to express this as a limitation.

Data analysis time span is not a requisite information for reporting. Could be removed from the text.

Results:

It would be good to add the country of origin of the respondents in table_1 if you have the data.

I think one of the most interesting results of your research is that the high dissatisfaction level of shifting pharmaceutical company (industry) sponsored meetings to online. This should be discussed more in discussion.

In the recommendations section, it would be better to cite relevant literature at the end of each item so that your contributions would be much clearer.

In the tables the presentation as “No. (%) is not a usual reporting style, should be rewritten as N (%)

Your manuscript would benefit from a careful proofreading related to English language usage.

Overall, I found your research interesting. But your manuscript needs some revisions which I stated above. In addition, I think that your manuscript is not contributing to the literature new or original results or conclusions.

I hope this would help to improve your manuscript.

Respectfully,

Reviewer #5: This manuscript evaluates a dramatic, sudden, and unexpected change in physician education. This is an important topic, not only for evaluation for current practices, but to consider the benefits future synergy between in-person and virtual learning. This manuscript proves some important evaluation of physician impressions of this transition, yet I believe it is somewhat limited in data that contribute to the conclusions provided. This could be improved by a more in-depth description of similar publications and careful consideration of the conclusions presented.

Introduction:

Some statements in the introduction seemed to bias the reader to the author's perspective and were opinion-based instead of facts. For example: "physicians have been bombarded with scientific webinars like no other time" and "Although at first, webinars seemed like a novelty, and probably the only solution for continuous medical education, however, months into the pandemic, these online meetings have lost much of their initial charm."

More of a review on previous research would be more appropriate and put this study in context. The authors state "There are limited data in the literature regarding physicians’ attitude and satisfaction towards this sudden shift in medical education" yet don't summarize any specific other publications at all in the introduction.

Methods:

Were any of the questions based on similar studies?

A complete list of the questions and possible responses would be helpful. Table 1 appears to be a partial list.

Results:

The question regarding "Attending more webinars" stated that the average of webinars observed this year had an average of 3 webinars; however, because the number of webinars observed last year was not given, it is unclear how much "more" is; this comparison seems crucial.

The authors discuss several types of virtual webinars, yet some questions seem to generalize. For example, for the question asking if webinars can replace in-person meetings, it seems like the answers may depend on the type of webinar offered (training courses, conferences, sponsored meetings). This is an example where having the full list of questions and responses could help interpretation.

Discussion:

Statistically speaking, as an overwhelming majority of responses were from the middle east, the discussion should focus on that population of physicians. There is not a broad enough representation across countries to make broader generalizations.

In describing % response comparisons of groups, providing a p value would be helpful (i.e. "This satisfaction was higher with International conferences and teaching courses compared to pharmaceutically-sponsored meetings (74.5% vs 41.4%)."

In the paragraph describing reasons for dissatisfaction in webinars, it would be helpful to get a sense of the frequency of similar responses to convert anecdotal feedback into generalizable statements.

For the recommendations, it would be helpful to place each recommendation in the context of previous literature. For example, some recommendations (#3, 4, 5, 6, 7) do not directly follow from the data in this study.

6. PLOS authors have the option to publish the peer review history of their article (what does this mean?). If published, this will include your full peer review and any attached files.

Reviewer #1: No

Reviewer #2: **Yes: **Mitra Amini

Reviewer #3: No

Reviewer #4: No

Reviewer #5: No

---

## [Author Response · Author response to Decision Letter 0]

17 Feb 2021

Response to editor and reviewers’ comments 

Dear Editor

Thank you for evaluating our manuscript. We have read with great concern the comments raised by the editor and reviewers, and we are glad to address and respond to these comments accordingly. 

Editor’s comment: 

I wish to be sincere and say that of five reviewers, four presented significant issues, questioning whether the study as performed, deserves to be accepted for publication. I would like to emphasize one criticism in a positive way. The overwhelming majority of the responders were from Arab countries. Consider transforming your study into one that relates to this specific population

We would like to thank the Editor for positive criticism, and we agree with the Editor on this point. More than 90% of the responders were from Arab and Middle East countries. We agree that this can limit the generalizability of our results. We emphasized on this point in the abstract, manuscript and mentioned it in Limitations section as well. All the changes made to the original draft are highlighted in yellow. 

Reviewer’s comment: 

Reviewer #1: 

Presentation

I appreciate the amount of work the authors have undertaken in this endeavor and have a few suggestions and clarifications that I hope may add to the value of this submitted manuscript.

1. Abstract

• The title of the article is clear and describes the focus of the study.

• The abstract is well structured and describes the study.

2. Introduction, the rationale and overall impact of the review are not clear. Some parts of the introduction need greater references to the evidence base.

3. Methods, the method section is in detail but there is no flow and many relevant pieces of information are missing:

• For international readers, and for readers who may wish to cite this study, it might be helpful for the authors to provide a brief overview of training activities at this particular institution to better facilitate an understanding of the results and discussion, especially as it applies to other institutions.

• Sampling technique and sampling size was not mentioned

• What were the baseline characteristics of the physicians such as qualification, specialties etc.

• How the consent from the physicians was taken?

• Who designed the survey?

• How the validity of the instrument was checked and what was the reliability score?

• It seems the snowball technique was not effectively used. Authors need to review the technique and explain in detail such as how they contact the first group, from where they got other contacts, who provided further contacts, and so on. In the snowball technique researcher asks for contacts and then asks contacts for participation. I am really curious about the process; how many reminders were sent and what was the logic for closing the link on November 15?

• The significance level should be checked.

4. Results & discussion, well written but the discussion needs more relevant referencing.

5. Conclusion, is relevant but due to major problems with the methods, it is really difficult to comment. In the conclusion, it would be useful to put the ideas in the context of what has been found. What is novel from this study? Strangely the validity of the tool was not done. Also, I am confused with the reason of “time constraint”. This needs further explanation.

6. Recommendations, are too long and most of them are not based on the results. Please review it.

7. References, Vancouver style was not followed.

8. It is unclear the additional value of this manuscript contributes to the literature beyond just a description of an attempt at investigating the physicians’ attitude toward webinars. The authors may need to highlight the values/implications of the current study (In the Introduction sections).

I also suggest careful editing of the manuscript to ensure clarity.

Response to Reviewer 1:

 1-We would like to thank Reviewer 1 for his appreciation, and we are glad to respond to the comments accordingly. 

2- We updated the Introduction section with the needed references. 

3- (a) As regard to the training activities for our institution, it included general online webinars in neurology, teaching courses on specific subjects (e.g. Botox injection), International Neurology conferences, and pharmaceutically sponsored meetings discussing specific therapies. However, the survey was distributed to other specialties which might has some differences. We added this part to the Methods section as per your recommendations. 

4- (b) As regard to the “snow-ball technique” used, the survey was initially distributed to the contacts of the primary investigators, and posted on personal social media accounts of the authors. Then it was posted on several and medical social media group. Responders were instructed to further distribute the survey to their contacts. The survey closed when there were no more responses for 1 day. This was mentioned in the Methods section (Line 166-167). 

5- (c) Baseline characteristics of the physicians were mentioned in the Results section, table 1, and table 2. It included demographic variables, specialties, and years of experience in their fields (instead of specific degrees, as this can be different in between countries). 

 (d) Accepting to fill the survey was considered an implied consent to participate in the study, as approved by the Institutional Review Board Committee, based on the fact that completing the survey was voluntary, and all answers were confidential. This is mentioned in Methods section. 

(e) The survey was designed by the authors of the study, and tested on a pilot of 25 physicians before distribution to physicians. The survey was not validated, and this was mentioned in the limitations section. 

 (f) Significance of the obtained results was judged at the 5% level.

 Thank you for amending the Results & Discussion. We updated the Discussion section with more relevant references as requested. 

5- We agree with the reviewer comment and Conclusion section was updated accordingly. 

6- Recommendations were equally based on the findings from responses from our survey, and we augmented this with a review of the current literature, as mentioned in the manuscript. It was formulated into points for simplicity and clarity. It can be further shortened if needed. 

7- We updated the references in Vancouver Style as needed. 

8- We believe this study is important in several aspects. First, this sudden change in education is new to physicians, and the number and frequency of webinars was found to be overwhelming to the majority of physicians. Second, we believe this study highlights such problem, and suggests further recommendations to regulate this “new-normal” practice. We further highlighted this in the Introduction section. 

Reviewer #2: 

Dear authors

Thank you for submitting your manuscript to the PLOS ONE journal. Your study is a cross-sectional study using a questionnaire. I think this manuscript is suitable for a letter to the editor. There is no novelty in the idea or the design.

Response to Reviewer 2:

We would like to thank Reviewer 2 for the interest in our cross-sectional study. The format suitable for publication is different among journals, and is decided by the journal’s Editor. We believe that the study is descriptive to an important topic, in addition to the suggested recommendations, which needs to be highlighted in the current format. 

Reviewer #3: 

The overall outcomes of this study are interesting and highlights the feelings of physicians on using online methods for CME etc. However, I feel that in parts it is superficial and needs re-written.

To me there needs to be greater clarity in how the results are being presented and discussed, with more cited comparisons (where appropriate) of the specific points.

I also think that some interesting points that the survey has identified have been overlooked and would have made for further discussion.

In the main the recommendations are good but follow a common-sense approach that many institutions already expect.

Given the International nature of medical education these days it is hard to select a time that suits all possible attendees around the world. The authors recommendation number 1 is therefore particularly difficult to address, unless it is time zone specific.

There is a good argument for having an institutional body to coordinate local CME activities, but again it is extremely difficult to coordinate if it is a webinar being presented by a specific professional society in another country. In the end it should be up to the physician to decide which webinar they want to attend or not.

Response to Reviewer 3:

We would like to thank Reviewer 3 for his interest in the study, and we are glad to respond to the comments accordingly. We agree with the Reviewer that some interesting points in the study needs further elucidation with more relevant references, and we updated the manuscript. As regard to the best timing, it should be regulated according to the institutional decision, however, the recommended time was based on responses from our responders and from literature review. However, we updated this point and mentioned that attending is a personal choice. 

Response to specific points:

Abstract: We updated the abstract after reading the full comments from reviewers on the manuscript.

Introduction -

Page 10. Line 19. What ‘proper regulation’ are the authors alluding to at this point in the paper?

As these changes in practice amid COVID-19 pandemic were sudden and unprecedented, we believe that “proper regulations” were needed as regard to the best teaching format, best timing to conduct webinars, proper method for feedback, ..etc. Most physicians needed to continue medical education while dealing with COVID-19 pandemic, and this was not possible with the number and frequency of webinars at the time. We updated the Introduction section with these important points. 

Page 10. Line 20. ‘several challenges and disadvantages have been reported by physicians.’ Is there a reference to this report?

A reference was added to the manuscript. 

Page 11. Line 10. Details of the ‘snowball technique’ should be provided.

Done. We explained more on this issue. 

Page 11. Line 17. How was a random sample of 25 physicians achieved? It is not clear how that was truly random

Each author sent the questionnaire to a random of 10 of his contacts, and 25 responded which was considered a convenience sample to this pilot. We updated the manuscript. Line 129

It is not clear why a distinction was drawn between international events and pharmaceutical sponsored events. this should be explained.

We thank the reviewer for this remark, and we believe this important issue should be explained more. International conferences are less frequent (usually annual), contain many webinars in a short period of time, not commercially biased, and eligible for higher CME accreditation, than pharmaceutically-sponsored events. We updated the manuscript accordingly. (Line 268-273)

Statistical analysis. I don’t think there is a need to include the dates that analysis was done.

We removed this line from the manuscript. 

Kolmogorov- Smirnov – should include ‘non-parametric test’

Done. The manuscript was updated accordingly

Results.

Why mention only female respondents and their age range? I think male respondents and their age range should also be included. That would make an interesting comparison. The way that the sentences 1 and 2 are written, suggests that the mean years of experience relates to the female respondents.

Done. We updated this part as requested. The mean age and mean years of experience were performed for the whole cohort. 

Page 14. Lines 9 and 10. ’Most of the respondents; 270 (82.8%) were mostly attendees’ or ‘15 (4.6%) were mostly speakers’ – using the term ‘mostly’ makes this unclear.

Because some respondents are both speakers and attendees, and some are only attendees, we agreed to add “mostly” to illustrate their major roles in the webinars they participate in. We put the words between brackets and in italics to remove this unclarity. 

Page 15. The figures presented in the first 2 paragraphs, indicate that respondents answered twice or more. Perhaps these results should be presented as a ranked order of preferences.

This is an important point. However, in the study we used a 5-point Likert scale of agreeableness ('strongly disagree', 'disagree', 'neutral', 'agree', and 'strongly agree’) against certain questions, which does not allow for ranking according to preference. Both figures (1) and (2) represent a stacked bar chart of these Likert questions. Two tables were also added to summarize the findings, and the survey questions were added as a supplementary file. (Appendix A). 

Page 15. Line 19. ‘that webinars and online meetings need further regulations’. If they need further regulation, what were the initial regulations?

Some regulations for online education and webinars in general are available:

- Guanci G. Best practices for webinars. Creative nursing. 2010 Aug 1;16(3):119-21.

- Mbati L, Minnaar A. Guidelines towards the facilitation of interactive online learning programmes in higher education. International Review of Research in Open and Distributed Learning. 2015;16(2):272-87.

However, no such studies were performed for the current situation following COVID-19 pandemic, and this was one of the main goals of this study. We updated this part in the manuscript. 

Discussion

Page 16. Lines 2 & 3. Again, results are being described as ‘mostly’.

We updated the manuscript for this point. 

Page 16. Line 20. ‘Moreover, the satisfaction with the scientific content was also higher between them (81.9% vs 36.8%)’. This statement needs to clarify between which groups.

Done. Line 268

Page 17. line 4. The authors again mention the need for further regulations, but there has been no mention of existing regulations.

We updated this point as stated in the Recommendations section. 

Page 18 line 1. It is not clear what the authors mean by poor communication, slow response time and increased distraction. I think these should be clarified.

We updated this point. Online education has less communication with attendees, slow response time with low Internet connection, increase distraction during webinars. 

Page 18. Line 5. Suggestion – ‘Furthermore, little evidence has been presented…..’

Done 

Recommendations.

Page 19. Line 6. ‘Studies have shown……’ Needs a reference here.

Page 19. Line 8. ‘Research in the field…..’ Needs a reference here.

In this section, the recommendations were pooled from the responses we had from the participating physicians, and from a review of the literature. It was then formulated into points for simplification and clarity. The literature review was derived from references number (25-32) for further reading. 

Page 20. Line 1. ‘The virtual experience should be enriched…’ It would be helpful if the authors gave examples of enrichment.

Done. We updated this point.

Conclusion.

The authors confuse the issue here. One the one hand they say ‘lacked proper regulation’ and on the other they say ‘the need for further regulation’, implying that there was some form of regulation in the first place.

We understand the confusion in this point. There have been several mentions of regulations for webinars in general in the literature. However, no specific recommendations were suggested for medical education during times of crisis or movement restrictions as in the current pandemic. We updated this point to remove such confusion and used the word “proper” throughout the manuscript. 

Reviewer #4: 

Dear respected Authors,

Thanks for your efforts on doing research and trying to contribute to the field of continuing medical education.

I have found your research question interesting and read your manuscript with interest.

Nonetheless, I have a few concerns regarding the manuscript which once addressed I believe would make the manuscript better.

Response to Reviewer 4:

We would like to thank Reviewer 4 for his interest in the study, and we are glad to respond to the comments accordingly. 

Abstract: Overall fine. It is better to rephrase “continuous medical education” to “continuing medical education (CME)”.

Done. We updated this part. 

Introduction:

The claims in the second paragraph should supported with relevant literature.

We agree and updated this section. 

In the purpose statement it has been stated that you aimed to assess “physicians’ attitude” which requires to use a valid reliable scale. Your aim could be defining the “physicians’ reactions” with the survey you developed.

We agree that a valid scale would have given more robust data. However, such scale assessing the desired outcome (attitude to medical education amid COVID-19 pandemic) is not available, to our knowledge. However, our scale was tested in a pilot study, and this was stated in the Methods and Limitations section. 

Method:

Your sampling method sounds more like convenience sampling rather than snowball sampling. I believe one of the major problems with your manuscript is the distribution of the survey questionnaire relatively uncontrolled way. Although you have distributed your survey questionnaire via multiple social media platforms, you have received vast majority of the responses from Middle East and North Africa region? Could it because of the survey questionnaire language? Was it in English or Arabic? This information should be included in the text.

We used a “snowball technique” in distributing the survey. The initial responders were from the contacts of the primary investigators who were mostly from the MENA region. However, responders were advised to recruit further participants through the snowball technique. The survey was also posted on several medical Facebook group, which also had majority of members from MENA region. We agree about this point and added it to Limitations section. 

What happened after the pilot study of the survey? Were there any changes or additions to the questionnaire?

Each author sent the questionnaire to a random of 10 of his contacts, and 25 responded, which was considered a convenient sample to this pilot. 

Yes, there had been some changes after the pilot mainly regarding formulation of questions, formulation of response preferences, and adding some demographic variables. 

Google Forms statement repeated multiple times in the manuscript which seems unnecessary.

Done, we updated the manuscript regarding this point. 

While it is a good measure to ask sign-in into Google accounts to avoid multiple responses from participants, this is also a limitation for “possible participants” who are not regularly using Google accounts or reluctant to log in. I think it would be better to express this as a limitation.

We had a discussion regarding this point before distributing the survey, and we agreed to accept only signed-in responses to have more credible data, and to avoid multiple responses. We thought of this as a point of strength actually, but if it is necessary to add it to limitations we can do this. 

Data analysis time span is not a requisite information for reporting. Could be removed from the text.

Done, we removed it. 

Results:

It would be good to add the country of origin of the respondents in table_1 if you have the data.

Unfortunately, this data is not available. We only collected the country of practice. 

I think one of the most interesting results of your research is that the high dissatisfaction level of shifting pharmaceutical company (industry) sponsored meetings to online. This should be discussed more in discussion.

We agree about this part. And we elucidated more on this in the discussion. Lines 268-273

In the recommendations section, it would be better to cite relevant literature at the end of each item so that your contributions would be much clearer.

In this section, the recommendations were pooled from the responses we had from the participating physicians, and from a review of the current literature. It was then formulated into points for simplification and clarity. The literature review was derived from references number (25-32) for future reading. 

In the tables the presentation as “No. (%) is not a usual reporting style, should be rewritten as N (%)

Done, we updated it. 

Your manuscript would benefit from a careful proofreading related to English language usage. Overall, I found your research interesting. But your manuscript needs some revisions which I stated above. In addition, I think that your manuscript is not contributing to the literature new or original results or conclusions.

I hope this would help to improve your manuscript.

Respectfully,

We thank the reviewer and we updated the manuscript as recommended. We believe physicians’ attitude towards online education is important in planning future educational activities, as for now, this sudden shift is unregulated and perceived by the majority as overwhelming. 

Reviewer #5: 

This manuscript evaluates a dramatic, sudden, and unexpected change in physician education. This is an important topic, not only for evaluation for current practices, but to consider the benefits future synergy between in-person and virtual learning. This manuscript proves some important evaluation of physician impressions of this transition, yet I believe it is somewhat limited in data that contribute to the conclusions provided. This could be improved by a more in-depth description of similar publications and careful consideration of the conclusions presented.

Response to Reviewer 5: 

We would like to thank Reviewer 5 for the interest and constructive remarks. We are glad to address the requested comments. 

Introduction:

Some statements in the introduction seemed to bias the reader to the author's perspective and were opinion-based instead of facts. For example: "physicians have been bombarded with scientific webinars like no other time" and "Although at first, webinars seemed like a novelty, and probably the only solution for continuous medical education, however, months into the pandemic, these online meetings have lost much of their initial charm."

More of a review on previous research would be more appropriate and put this study in context. The authors state "There are limited data in the literature regarding physicians’ attitude and satisfaction towards this sudden shift in medical education" yet don't summarize any specific other publications at all in the introduction.

We agree with the reviewer on this point, and updated the manuscript. The other publications were summarized in the Discussion section rather than Introduction, due to space limitations. However, we added the references to such publications in the Introduction section. (Ref 7 and 8) 

Methods:

Were any of the questions based on similar studies?

A complete list of the questions and possible responses would be helpful. Table 1 appears to be a partial list.

The questions were based on similar literature assessing physician’s attitudes towards certain outcomes. It was tested and remodified in a pilot study. We added the full list of the questions as a supplementary file. 

Results:

The question regarding "Attending more webinars" stated that the average of webinars observed this year had an average of 3 webinars; however, because the number of webinars observed last year was not given, it is unclear how much "more" is; this comparison seems crucial.

We agree that this information is crucial. Because the vast majority of respondents reported not attending webinars at all during the last year, but around 60% attended more meetings in general this year, the results were presented in such manner. We updated the manuscript as requested. 

The authors discuss several types of virtual webinars, yet some questions seem to generalize. For example, for the question asking if webinars can replace in-person meetings, it seems like the answers may depend on the type of webinar offered (training courses, conferences, sponsored meetings). This is an example where having the full list of questions and responses could help interpretation.

In this question, we aimed to have a general perception of this sudden shift in educational format, and if it can replace in-person traditional teaching, regardless of the type of webinars. The rationale is that this online format is relatively new to the medical community, and most educational activities used to be in-person. The attitude towards different types was evaluated in other questions. 

Discussion:

Statistically speaking, as an overwhelming majority of responses were from the middle east, the discussion should focus on that population of physicians. There is not a broad enough representation across countries to make broader generalizations.

We agree with the reviewer on this point. The majority of physicians were working in Arab and MENA countries. We focused more in the discussion about this point as requested, and also mentioned this in the Limitations part. 

In describing % response comparisons of groups, providing a p value would be helpful (i.e. "This satisfaction was higher with International conferences and teaching courses compared to pharmaceutically-sponsored meetings (74.5% vs 41.4%)."

We updated this part in Results and Discussion. New tables were added including these findings (Tables 3,4).

In the paragraph describing reasons for dissatisfaction in webinars, it would be helpful to get a sense of the frequency of similar responses to convert anecdotal feedback into generalizable statements.

We agree with this point and updated the manuscript accordingly. From line 294-300.

For the recommendations, it would be helpful to place each recommendation in the context of previous literature. For example, some recommendations (#3, 4, 5, 6, 7) do not directly follow from the data in this study.

In this section, the recommendations were pooled from the responses we had from the participating physicians, and from a review of the literature. It was then formulated into points for simplification and clarity. The literature review was derived from references number (25-32) for further reading.

---

## [Decision Letter · Decision Letter 1]

22 Mar 2021

PONE-D-20-39984R1

Physicians' attitude towards webinars and online education amid COVID-19 pandemic: When Less is More

PLOS ONE

Dear Dr. Ismail,

Thank you for submitting your manuscript to PLOS ONE. After careful consideration, we feel that it has merit but does not fully meet PLOS ONE’s publication criteria as it currently stands. Therefore, we invite you to submit a revised version of the manuscript that addresses the points raised during the review process.

We look forward to receiving your revised manuscript.

Kind regards,

Itamar Ashkenazi

Academic Editor

PLOS ONE

Journal Requirements:

Additional Editor Comments (if provided):

Reviewers' comments:

Reviewer's Responses to Questions

**Comments to the Author**

1. If the authors have adequately addressed your comments raised in a previous round of review and you feel that this manuscript is now acceptable for publication, you may indicate that here to bypass the “Comments to the Author” section, enter your conflict of interest statement in the “Confidential to Editor” section, and submit your "Accept" recommendation.

Reviewer #1: All comments have been addressed

Reviewer #4: (No Response)

Reviewer #5: (No Response)

2. Is the manuscript technically sound, and do the data support the conclusions?

Reviewer #1: Yes

Reviewer #4: Partly

Reviewer #5: Partly

3. Has the statistical analysis been performed appropriately and rigorously? 

Reviewer #1: Yes

Reviewer #4: Yes

Reviewer #5: No

4. Have the authors made all data underlying the findings in their manuscript fully available?

Reviewer #1: Yes

Reviewer #4: Yes

Reviewer #5: Yes

5. Is the manuscript presented in an intelligible fashion and written in standard English?

Reviewer #1: Yes

Reviewer #4: No

Reviewer #5: Yes

6. Review Comments to the Author

Reviewer #1: (No Response)

Reviewer #4: Dear Respected Authors,

Thanks for your efforts trying to improve your research manuscript according to the reviewers' comments and suggestions.

Although you have mentioned you agree that a valid scale needed to be able to measure the "attitude" of physicians, you did not made any changes or improvements in neither title nor text of your manuscript.

Pilot study is still unclear in terms of method and findings in the manuscript.

Best regards

Reviewer #5: Thank you for your resubmission of your publication. I have several comments, mostly relating to my initial evaluation. One new comment is in regards to your statistical analysis in Table 3 and 4. I am not sure what your comparisons are for each question. You used a Goodness of Fit, but I'm not sure what your comparison values were. Any statistical analysis on these responses doesn't seem necessary.

Regarding your survey population, I believe the response group should be noted in the title or abstract.

For your recommendations at the end of the paper, it would be helpful if you referenced the other publications that recommendations not directly related to your study are from. You mention that some of the recommendations are from feedback in your survey; could you add the specific text they provided? (ie. Recommendation 2 does not cite the publication for preference for Tuesday; Recommendation 3 mentions previous work on seminar length but does not cite it; 5-7 seem to be more opinions)

7. PLOS authors have the option to publish the peer review history of their article (what does this mean?). If published, this will include your full peer review and any attached files.

Reviewer #1: No

Reviewer #4: No

Reviewer #5: No

---

## [Author Response · Author response to Decision Letter 1]

23 Mar 2021

Review Comments to the Author

Reviewer #1: (No Response)

Reviewer #4: 

Dear Respected Authors,

Thanks for your efforts trying to improve your research manuscript according to the reviewers' comments and suggestions.

Although you have mentioned you agree that a valid scale needed to be able to measure the "attitude" of physicians, you did not made any changes or improvements in neither title nor text of your manuscript.

We would like to thank the respected reviewer for the constructive remarks. We agree with the reviewer that our survey was not validated, and this was mentioned in the Limitations section. We used 5-point Likert scale of agreeableness which has been used in several studies to measure attitude towards different variables. We updated the limitations section as regard to the reliability in measuring attitude. 

Pilot study is still unclear in terms of method and findings in the manuscript.

Each author sent the questionnaire to a random of 10 physicians of his contacts, and 25 responded, which was considered a convenient sample to this pilot. The Methods section was updated (Line 129-134), highlighted in green. 

Best regards

Reviewer #5: 

Thank you for your resubmission of your publication. I have several comments, mostly relating to my initial evaluation. One new comment is in regards to your statistical analysis in Table 3 and 4. I am not sure what your comparisons are for each question. You used a Goodness of Fit, but I'm not sure what your comparison values were. Any statistical analysis on these responses doesn't seem necessary.

We would like to thank the reviewer for the constructive remarks. These tables were added upon your suggestions to add (p value) in describing the responses. However, we can remove them and keep the p values only, if this is recommended and acceptable by the journal editor. 

Regarding your survey population, I believe the response group should be noted in the title or abstract.

We added this point to the abstract (highlighted in green) as suggested. We focused on this point in the Discussion as requested. Also, this was mentioned in the Limitations part.

For your recommendations at the end of the paper, it would be helpful if you referenced the other publications that recommendations not directly related to your study are from. You mention that some of the recommendations are from feedback in your survey; could you add the specific text they provided? (ie. Recommendation 2 does not cite the publication for preference for Tuesday; Recommendation 3 mentions previous work on seminar length but does not cite it; 5-7 seem to be more opinions)

We updated this part in the manuscript and assigned the references used for these recommendations.

---

## [Editor Report · Decision Letter 2]

5 Apr 2021

Physicians' attitude towards webinars and online education amid COVID-19 pandemic: When Less is More

PONE-D-20-39984R2

Dear Dr. Ismail,

We’re pleased to inform you that your manuscript has been judged scientifically suitable for publication and will be formally accepted for publication once it meets all outstanding technical requirements.

Kind regards,

Itamar Ashkenazi

Academic Editor

PLOS ONE
---

## [Editor Report · Acceptance letter]

8 Apr 2021

PONE-D-20-39984R2 

Physicians' attitude towards webinars and online education amid COVID-19 pandemic: When Less is More 

Dear Dr. Ismail:

I'm pleased to inform you that your manuscript has been deemed suitable for publication in PLOS ONE. Congratulations! Your manuscript is now with our production department. 

Kind regards, 

on behalf of

Dr. Itamar Ashkenazi 

Academic Editor

PLOS ONE